# Augmented Neural ODEs

**Emilien Dupont**
University of Oxford
dupont@stats.ox.ac.uk

**Arnaud Doucet**
University of Oxford
doucet@stats.ox.ac.uk

**Yee Whye Teh**
University of Oxford
y.w.teh@stats.ox.ac.uk

## Abstract

We show that Neural Ordinary Differential Equations (ODEs) learn representations that preserve the topology of the input space and prove that this implies the existence of functions Neural ODEs cannot represent. To address these limitations, we introduce Augmented Neural ODEs which, in addition to being more expressive models, are empirically more stable, generalize better and have a lower computational cost than Neural ODEs.

## 1   Introduction

The relationship between neural networks and differential equations has been studied in several recent works (Weinan, 2017; Lu et al., 2017; Haber & Ruthotto, 2017; Ruthotto & Haber, 2018; Chen et al., 2018). In particular, it has been shown that Residual Networks (He et al., 2016) can be interpreted as discretized ODEs. Taking the discretization step to zero gives rise to a family of models called Neural ODEs (Chen et al., 2018). These models can be efficiently trained with backpropagation and have shown great promise on a number of tasks including modeling continuous time data and building normalizing flows with low computational cost (Chen et al., 2018; Grathwohl et al., 2018).

In this work, we explore some of the consequences of taking this continuous limit and the restrictions this might create compared with regular neural nets. In particular, we show that there are simple classes of functions Neural ODEs (NODEs) cannot represent. While it is often possible for NODEs to approximate these functions in practice, the resulting flows are complex and lead to ODE problems that are computationally expensive to solve. To overcome these limitations, we introduce Augmented Neural ODEs (ANODEs) which are a simple extension of NODEs. ANODEs augment the space on which the ODE is solved, allowing the model to use the additional dimensions to learn more complex functions using simpler flows (see Fig. 1). In addition to being more expressive models, ANODEs significantly reduce the computational cost of both forward and backward passes of the model compared with NODEs. Our experiments also show that ANODEs generalize better, achieve lower losses with fewer parameters and are more stable to train.

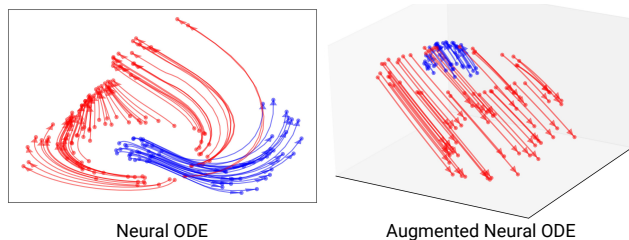

Neural ODE          Augmented Neural ODE

Figure 1: Learned flows for a Neural ODE and an Augmented Neural ODE. The flows (shown as lines with arrows) map input points to linearly separable features for binary classification. Augmented Neural ODEs learn simpler flows that are easier for the ODE solver to compute.

## 2 Neural ODEs

NODEs are a family of deep neural network models that can be interpreted as a continuous equivalent of Residual Networks (ResNets). To see this, consider the transformation of a hidden state from a layer $t$ to $t+1$ in ResNets

$$\mathbf{h}_{t+1} = \mathbf{h}_t + \mathbf{f}_t(\mathbf{h}_t)$$

where $\mathbf{h}_t \in \mathbb{R}^d$ is the hidden state at layer $t$ and $\mathbf{f}_t : \mathbb{R}^d \to \mathbb{R}^d$ is some differentiable function which preserves the dimension of $\mathbf{h}_t$ (typically a CNN). The difference $\mathbf{h}_{t+1} - \mathbf{h}_t$ can be interpreted as a discretization of the derivative $\mathbf{h}'(t)$ with timestep $\Delta t = 1$. Letting $\Delta t \to 0$, we see that

$$\lim_{\Delta t \to 0} \frac{\mathbf{h}_{t+\Delta t} - \mathbf{h}_t}{\Delta t} = \frac{\mathrm{d}\mathbf{h}(t)}{\mathrm{d}t} = \mathbf{f}(\mathbf{h}(t), t)$$

so the hidden state can be parameterized by an ODE. We can then map a data point $\mathbf{x}$ into a set of features $\phi(\mathbf{x})$ by solving the Initial Value Problem (IVP)

$$\frac{\mathrm{d}\mathbf{h}(t)}{\mathrm{d}t} = \mathbf{f}(\mathbf{h}(t), t), \qquad \mathbf{h}(0) = \mathbf{x}$$

to some time $T$. The hidden state at time $T$, i.e. $\mathbf{h}(T)$, corresponds to the features learned by the model. The analogy with ResNets can then be made more explicit. In ResNets, we map an input $\mathbf{x}$ to some output $\mathbf{y}$ by a forward pass of the neural network. We then adjust the weights of the network to match $\mathbf{y}$ with some $\mathbf{y}_{\text{true}}$. In NODEs, we map an input $\mathbf{x}$ to an output $\mathbf{y}$ by solving an ODE starting from $\mathbf{x}$. We then adjust the dynamics of the system (encoded by $\mathbf{f}$) such that the ODE transforms $\mathbf{x}$ to a $\mathbf{y}$ which is close to $\mathbf{y}_{\text{true}}$.

**ODE flows.** We also define the flow associated to the vector field $\mathbf{f}(\mathbf{h}(t), t)$ of the ODE. The flow $\phi_t : \mathbb{R}^d \to \mathbb{R}^d$ is defined as the hidden state at time $t$, i.e. $\phi_t(\mathbf{x}) = \mathbf{h}(t)$, when solving the ODE from the initial condition $\mathbf{h}(0) = \mathbf{x}$. The flow measures how the states of the ODE at a given time $t$ depend on the initial conditions $\mathbf{x}$. We define the features of the ODE as $\phi(\mathbf{x}) := \phi_T(\mathbf{x})$, i.e. the flow at the final time $T$ to which we solve the ODE.

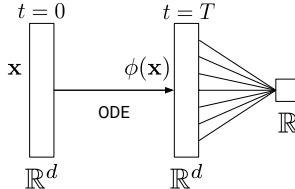

Figure 2: Diagram of Neural ODE architecture.

**NODEs for regression and classification.** We can use ODEs to map input data $\mathbf{x} \in \mathbb{R}^d$ to a set of features or representations $\phi(\mathbf{x}) \in \mathbb{R}^d$. However, we are often interested in learning functions from $\mathbb{R}^d$ to $\mathbb{R}$, e.g. for regression or classification. To define a model from $\mathbb{R}^d$ to $\mathbb{R}$, we follow the example given in Lin & Jegelka (2018) for ResNets. We define the NODE $g : \mathbb{R}^d \to \mathbb{R}$ as $g(\mathbf{x}) = \mathcal{L}(\phi(\mathbf{x}))$ where $\mathcal{L} : \mathbb{R}^d \to \mathbb{R}$ is a linear map and $\phi : \mathbb{R}^d \to \mathbb{R}^d$ is the mapping from data to features. As shown in Fig. 2, this is a simple model architecture: an ODE layer, followed by a linear layer.

## 3 A simple example in 1d

In this section, we introduce a simple function that ODE flows cannot represent, motivating many of the examples discussed later. Let $g_{1d} : \mathbb{R} \to \mathbb{R}$ be a function such that $g_{1d}(-1) = 1$ and $g_{1d}(1) = -1$.

**Proposition 1.** *The flow of an ODE cannot represent $g_{1d}(x)$.*

A detailed proof is given in the appendix. The intuition behind the proof is simple; the trajectories mapping $-1$ to $1$ and $1$ to $-1$ must intersect each other (see Fig. 3). However, ODE trajectories cannot cross each other, so the flow of an ODE cannot represent $g_{1d}(x)$. This simple observation is at the core of all the examples provided in this paper and forms the basis for many of the limitations of NODEs.

**Experiments.** We verify this behavior experimentally by training an ODE flow on the identity mapping and on $g_{1d}(x)$. The resulting flows are shown in Fig. 3. As can be seen, the model easily learns the identity mapping but cannot represent $g_{1d}(x)$. Indeed, since the trajectories cannot cross, the model maps all input points to zero to minimize the mean squared error.

**ResNets vs NODEs.** NODEs can be interpreted as continuous equivalents of ResNets, so it is interesting to consider why ResNets can represent $g_{1d}(x)$ but NODEs cannot. The reason for this is

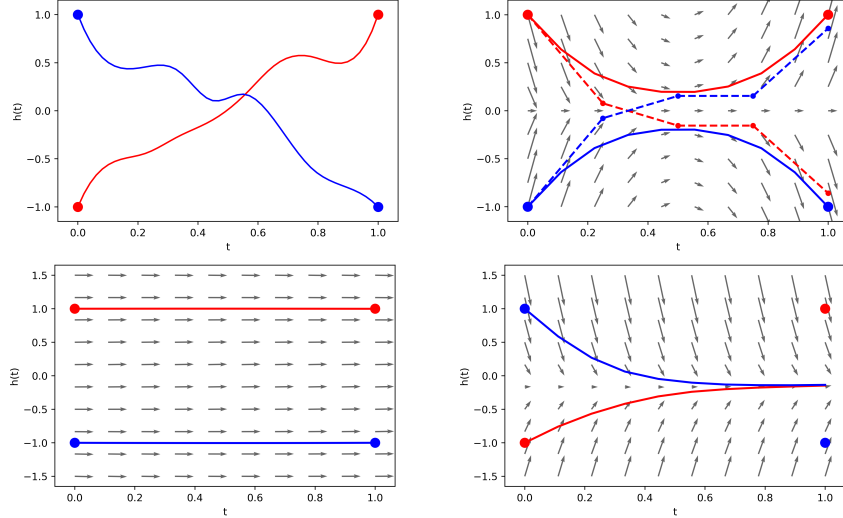

Figure 3: (Top left) Continuous trajectories mapping $-1$ to $1$ (red) and $1$ to $-1$ (blue) must intersect each other, which is not possible for an ODE. (Top right) Solutions of the ODE are shown in solid lines and solutions using the Euler method (which corresponds to ResNets) are shown in dashed lines. As can be seen, the discretization error allows the trajectories to cross. (Bottom) Resulting vector fields and trajectories from training on the identity function (left) and $g_{1\mathrm{d}}(x)$ (right).

exactly because ResNets are a discretization of the ODE, allowing the trajectories to make discrete jumps to cross each other (see Fig. 3). Indeed, the error arising when taking discrete steps allows the ResNet trajectories to cross. In this sense, ResNets can be interpreted as ODE solutions with large errors, with these errors allowing them to represent more functions.

## 4 Functions Neural ODEs cannot represent

We now introduce classes of functions in arbitrary dimension $d$ which NODEs cannot represent. Let $0 < r_1 < r_2 < r_3$ and let $g : \mathbb{R}^d \to \mathbb{R}$ be a function such that

$$\begin{cases} g(\mathbf{x}) = -1 & \text{if } \|\mathbf{x}\| \leq r_1 \\ g(\mathbf{x}) = 1 & \text{if } r_2 \leq \|\mathbf{x}\| \leq r_3, \end{cases}$$

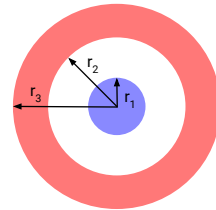

where $\| \cdot \|$ is the Euclidean norm. An illustration of this function for $d = 2$ is shown in Fig. 4. The function maps all points inside the blue sphere to $-1$ and all points in the red annulus to $1$.

Figure 4: Diagram of $g(\mathbf{x})$ for $d = 2$.

**Proposition 2.** *Neural ODEs cannot represent $g(\mathbf{x})$.*

A proof is given in the appendix. While the proof requires tools from ODE theory and topology, the intuition behind it is simple. In order for the linear layer to map the blue and red points to $-1$ and $1$ respectively, the features $\phi(\mathbf{x})$ for the blue and red points must be linearly separable. Since the blue region is enclosed by the red region, points in the blue region must cross over the red region to become linearly separable, requiring the trajectories to intersect, which is not possible. In fact, we can make more general statements about which features Neural ODEs can learn.

**Proposition 3.** *The feature mapping $\phi(\mathbf{x})$ is a homeomorphism, so the features of Neural ODEs preserve the topology of the input space.*

A proof is given in the appendix. This statement is a consequence of the flow of an ODE being a homeomorphism, i.e. a continuous bijection whose inverse is also continuous; see, e.g., (Younes, 2010). This implies that NODEs can only continuously deform the input space and cannot for example tear a connected region apart.

**Discrete points and continuous regions.** It is worthwhile to consider what these results mean in practice. Indeed, when optimizing NODEs we train on inputs which are sampled from the continuous regions of the annulus and the sphere (see Fig. 4). The flow could then squeeze through the gaps

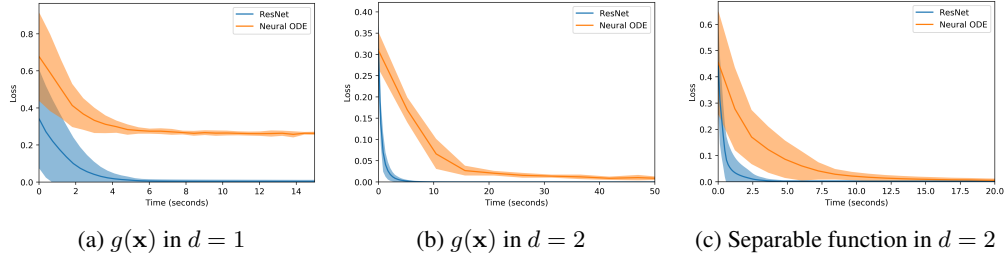

(a) $g(\mathbf{x})$ in $d = 1$        (b) $g(\mathbf{x})$ in $d = 2$        (c) Separable function in $d = 2$

Figure 5: Comparison of training losses of NODEs and ResNets. Compared to ResNets, NODEs struggle to fit $g(\mathbf{x})$ both in $d = 1$ and $d = 2$. The difference between ResNets and NODEs is less pronounced for the separable function.

between sampled points making it possible for the NODE to learn a good approximation of the function. However, flows that need to stretch and squeeze the input space in such a way are likely to lead to ill-posed ODE problems that are numerically expensive to solve. In order to explore this, we run a number of experiments (the code to reproduce all experiments in this paper is available at `https://github.com/EmilienDupont/augmented-neural-odes`).

## 4.1 Experiments

We first compare the performance of ResNets and NODEs on simple regression tasks. To provide a baseline, we not only train on $g(\mathbf{x})$ but also on data which can be made linearly separable without altering the topology of the space (implying that Neural ODEs should be able to easily learn this function). To ensure a fair comparison, we run large hyperparameter searches for each model and repeat each experiment 20 times to ensure results are meaningful across initializations (see appendix for details). We show results for experiments with $d = 1$ and $d = 2$ in Fig. 5. For $d = 1$, the ResNet easily fits the function, while the NODE cannot approximate $g(\mathbf{x})$. For $d = 2$, the NODE eventually learns to approximate $g(\mathbf{x})$, but struggles compared to ResNets. This problem is less severe for the separable function, presumably because the flow does not need to break apart any regions to linearly separate them.

## 4.2 Computational Cost and Number of Function Evaluations

One of the known limitations of NODEs is that, as training progresses and the flow gets increasingly complex, the number of steps required to solve the ODE increases (Chen et al., 2018; Grathwohl et al., 2018). As the ODE solver evaluates the function $\mathbf{f}$ at each step, this problem is often referred to as the increasing number of function evaluations (NFE). In Fig. 6, we visualize the evolution of the feature space during training and the corresponding NFEs. The NODE initially tries to move the inner sphere out of the annulus by pushing against and stretching the barrier. Eventually, since we are mapping discrete points and not a continuous region, the flow is able to break apart the annulus to let the flow through. However, this results in a large increase in NFEs, implying that the ODE stretching the space to separate the two regions becomes more difficult to solve, making the computation slower.

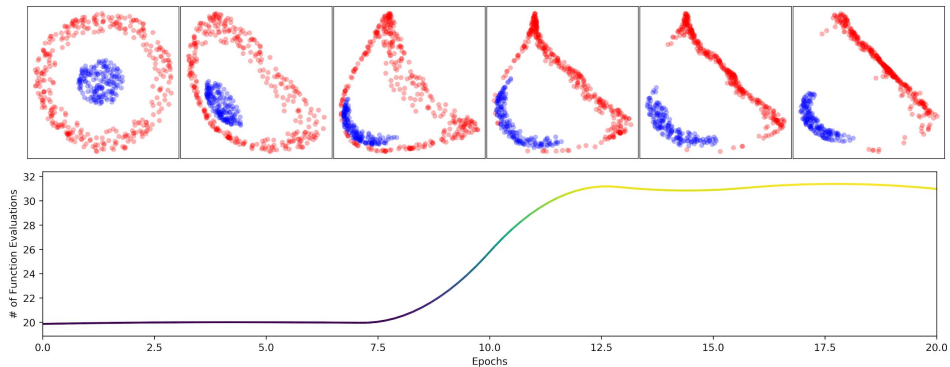

Figure 6: Evolution of the feature space as training progresses and the corresponding number of function evaluations required to solve the ODE. As the ODE needs to break apart the annulus, the number of function evaluations increases.

# 5 Augmented Neural ODEs

Motivated by our theory and experiments, we introduce Augmented Neural ODEs (ANODEs) which provide a simple solution to the problems we have discussed. We augment the space on which we learn and solve the ODE from $\mathbb{R}^d$ to $\mathbb{R}^{d+p}$, allowing the ODE flow to lift points into the additional dimensions to avoid trajectories intersecting each other. Letting $\mathbf{a}(t) \in \mathbb{R}^p$ denote a point in the augmented part of the space, we can formulate the augmented ODE problem as

$$\frac{\mathrm{d}}{\mathrm{d}t} \begin{bmatrix} \mathbf{h}(t) \\ \mathbf{a}(t) \end{bmatrix} = \mathbf{f}( \begin{bmatrix} \mathbf{h}(t) \\ \mathbf{a}(t) \end{bmatrix}, t), \qquad \begin{bmatrix} \mathbf{h}(0) \\ \mathbf{a}(0) \end{bmatrix} = \begin{bmatrix} \mathbf{x} \\ \mathbf{0} \end{bmatrix}$$

i.e. we concatenate every data point $\mathbf{x}$ with a vector of zeros and solve the ODE on this augmented space. We hypothesize that this will also make the learned (augmented) $\mathbf{f}$ smoother, giving rise to simpler flows that the ODE solver can compute in fewer steps. In the following sections, we verify this behavior experimentally and show both on toy and image datasets that ANODEs achieve lower losses, better generalization and lower computational cost than regular NODEs.

## 5.1 Experiments

We first compare the performance of NODEs and ANODEs on toy datasets. As in previous experiments, we run large hyperparameter searches to ensure a fair comparison. As can be seen on Fig. 7, when trained on $g(\mathbf{x})$ in different dimensions, ANODEs are able to fit the functions NODEs cannot and learn much faster than NODEs despite the increased dimension of the input. The corresponding flows learned by the model are shown in Fig. 7. As can be seen, in $d = 1$, the ANODE moves into a higher dimension to linearly separate the points, resulting in a simple, nearly linear flow. Similarly, in $d = 2$, the NODE learns a complicated flow whereas ANODEs simply lift out the inner circle to separate the data. This effect can also be visualized as the features evolve during training (see Fig. 8).

**Computational cost and number of function evaluations.** As ANODEs learn simpler flows, they would presumably require fewer iterations to compute. To test this, we measure the NFEs for NODEs and ANODEs when training on $g(\mathbf{x})$. As can be seen in Fig. 8, the NFEs required by ANODEs hardly increases during training while it nearly doubles for NODEs. We obtain similar results when training NODEs and ANODEs on image datasets (see Section 5.2).

**Generalization.** As ANODEs learn simpler flows, we also hypothesize that they generalize better to unseen data than NODEs. To test this, we first visualize to which value each point in the input space gets mapped by a NODE and an ANODE that have been optimized to approximately zero training loss. As can be seen in Fig. 9, since NODEs can only continuously deform the input space, the learned flow must squeeze the points in the inner circle through the annulus, leading to poor

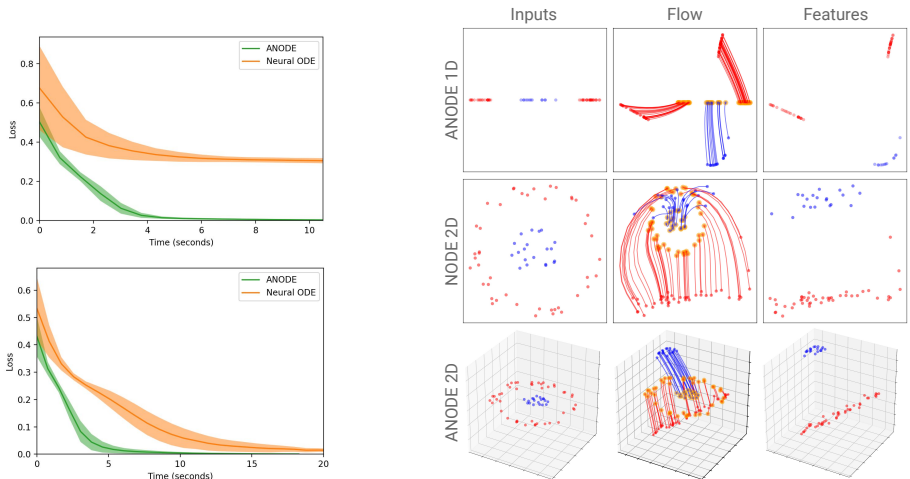

Figure 7: (Left) Loss plots for NODEs and ANODEs trained on $g(\mathbf{x})$ in $d = 1$ (top) and $d = 2$ (bottom). ANODEs easily approximate the functions and are consistently faster than NODEs. (Right) Flows learned by NODEs and ANODEs. ANODEs learn simple nearly linear flows while NODEs learn complex flows that are difficult for the ODE solver to compute.

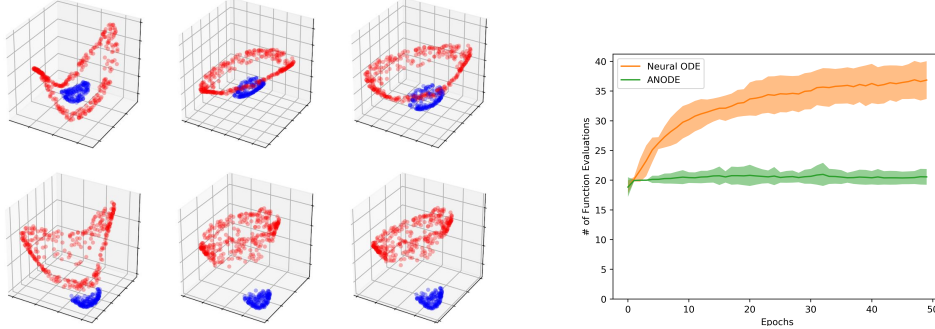

Figure 8: (Left) Evolution of features during training for ANODEs. The top left tile shows the feature space for a randomly initialized ANODE and the bottom right tile shows the features after training. (Right) Evolution of the NFEs during training for NODEs and ANODEs trained on $g(\mathbf{x})$ in $d = 1$.

generalization. ANODEs, in contrast, map all points in the input space to reasonable values. As a further test, we can also create a validation set by removing random slices of the input space (e.g. removing all points whose angle is in $[0, \frac{\pi}{5}]$) from the training set. We train both NODEs and ANODEs on the training set and plot the evolution of the validation loss during training in Fig. 9. While there is a large generalization gap for NODEs, presumably because the flow moves through the gaps in the training set, ANODEs generalize much better and achieve near zero validation loss.

As we have shown, experimentally we obtain lower losses, simpler flows, better generalization and ODEs requiring fewer NFEs to solve when using ANODEs. We now test this behavior on image data by training models on MNIST, CIFAR10, SVHN and 200 classes of $64 \times 64$ ImageNet.

## 5.2 Image Experiments

We perform experiments on image datasets using convolutional architectures for $\mathbf{f}(\mathbf{h}(t), t)$. As the input $\mathbf{x}$ is an image, the hidden state $\mathbf{h}(t)$ is now in $\mathbb{R}^{c \times h \times w}$ where $c$ is the number of channels and $h$ and $w$ are the height and width respectively. In the case where $\mathbf{h}(t) \in \mathbb{R}^d$ we augmented the space as $\mathbf{h}(t) \in \mathbb{R}^{d+p}$. For images we augment the space as $\mathbb{R}^{c \times h \times w} \to \mathbb{R}^{(c+p) \times h \times w}$, i.e. we add $p$ channels of zeros to the input image. While there are other ways to augment the space, we found that increasing the number of channels works well in practice and use this method for all experiments. Full training and architecture details can be found in the appendix.

Results for models trained with and without augmentation are shown in Fig. 10. As can be seen, ANODEs train faster and obtain lower losses at a smaller computational cost than NODEs. On MNIST for example, ANODEs with 10 augmented channels achieve the same loss in roughly 10 times fewer iterations (for CIFAR10, ANODEs are roughly 5 times faster). Perhaps most interestingly, we can plot the NFEs against the loss to understand roughly how complex a flow (i.e. how many NFEs) are required to model a function that achieves a certain loss. For example, to compute a function which obtains a loss of 0.8 on CIFAR10, a NODE requires approximately 100 function evaluations whereas ANODEs only require 50. Similar observations can be made for other datasets, implying that ANODEs can model equally rich functions at half the computational cost of NODEs.

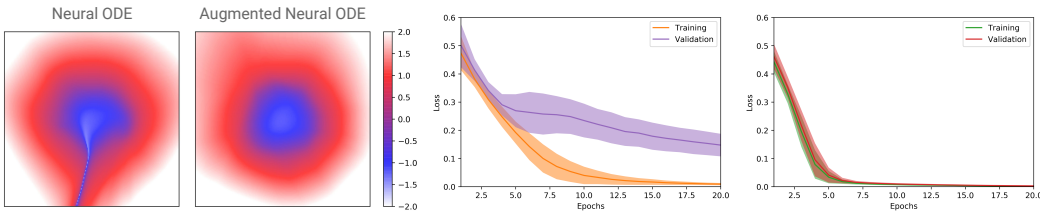

Figure 9: (Left) Plots of how NODEs and ANODEs map points in the input space to different outputs (both models achieve approximately the same zero training loss). As can be seen, the ANODE generalizes better. (Middle) Training and validation losses for NODE. (Right) Training and validation losses for ANODE.

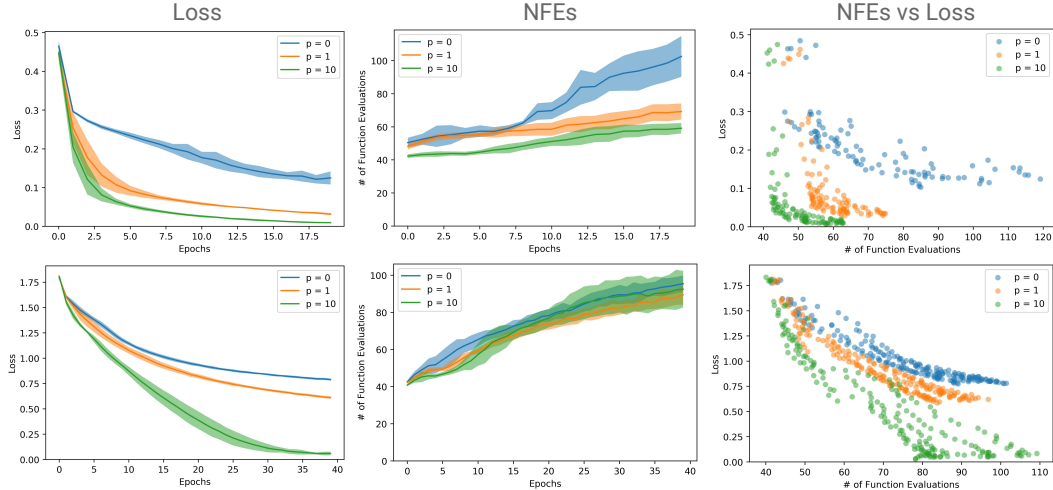

Figure 10: Training losses, NFEs and NFEs vs Loss for various augmented models on MNIST (top row) and CIFAR10 (bottom row). Note that $p$ indicates the size of the augmented dimension, so $p = 0$ corresponds to a regular NODE model. Further results on SVHN and $64 \times 64$ ImageNet can be found in the appendix.

**Parameter efficiency.** As we augment the dimension of the ODEs, we also increase the number of parameters of the models, so it may be that the improved performance of ANODEs is due to the higher number of parameters. To test this, we train a NODE and an ANODE with the same number of parameters on MNIST (84k weights), CIFAR10 (172k weights), SVHN (172k weights) and $64 \times 64$ ImageNet (366k weights). We find that the augmented model achieves significantly lower losses with fewer NFEs than the NODE, suggesting that ANODEs use the parameters more efficiently than NODEs (see appendix for details and results). For all subsequent experiments, we use NODEs and ANODEs with the same number of parameters.

**NFEs and weight decay.** The increased computational cost during training is a known issue with NODEs and has previously been tackled by adding weight decay (Grathwohl et al., 2018). As ANODEs also achieve lower computational cost, we test models with various combinations of weight decay and augmentation (see appendix for detailed results). We find that ANODEs without weight decay significantly outperform NODEs with weight decay. However, using both weight decay and augmentation achieves the lowest NFEs at the cost of a slightly higher loss. Combining augmentation with weight decay may therefore be a fruitful avenue for further scaling NODE models.

**Accuracy.** Fig. 11 shows training accuracy against NFEs for ANODEs and NODEs on MNIST, CIFAR10 and SVHN. As expected, ANODEs achieve higher accuracy at a lower computational cost than NODEs (similar results hold for ImageNet as shown in the appendix).

**Generalization for images.** As noted in Section 5.1, ANODEs generalize better than NODEs on simple datasets, presumably because they learn simpler and smoother flows. We also test this behavior on image datasets by training models with and without augmentation on the training set and calculating the loss and accuracy on the test set. As can be seen in Fig. 12 and Table 1, for MNIST, CIFAR10, and SVHN, ANODEs achieve lower test losses and higher test accuracies than NODEs, suggesting that ANODEs also generalize better on image datasets.

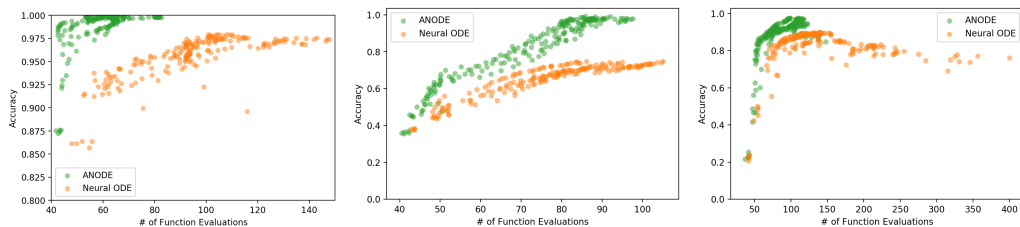

Figure 11: Accuracy vs NFEs for MNIST (left), CIFAR10 (middle) and SVHN (right).

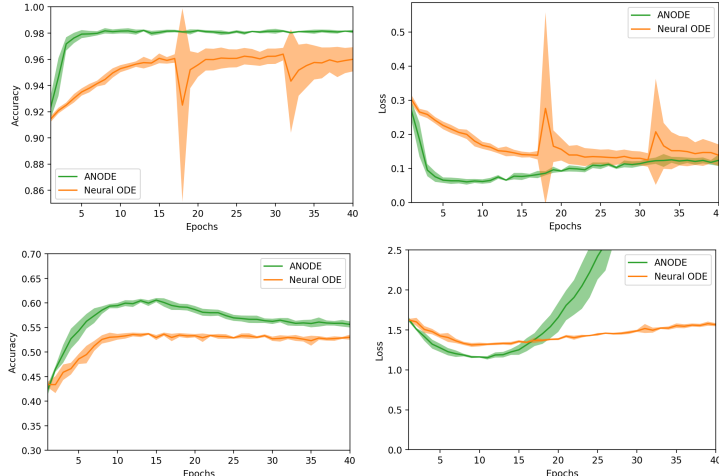

Figure 12: Test accuracy (left) and loss (right) for MNIST (top) and CIFAR10 (bottom).

|         | NODE              | ANODE                |
|---------|-------------------|----------------------|
| MNIST   | $96.4\% \pm 0.5$  | $\mathbf{98.2\% \pm 0.1}$ |
| CIFAR10 | $53.7\% \pm 0.2$  | $\mathbf{60.6\% \pm 0.4}$ |
| SVHN    | $81.0\% \pm 0.6$  | $\mathbf{83.5\% \pm 0.5}$ |

Table 1: Test accuracies and their standard deviation over 5 runs on various image datasets.

**Stability.** While experimenting with NODEs we found that the NFEs could often become prohibitively large (in excess of 1000, which roughly corresponds to a 1000-layer ResNet). For example, when overfitting a NODE on MNIST, the learned flow can become so ill posed the ODE solver requires timesteps that are smaller than machine precision resulting in underflow. Further, this complex flow often leads to unstable training resulting in exploding losses. As shown in Fig. 13, augmentation consistently leads to stable training and fewer NFEs, even when overfitting.

**Scaling.** To measure how well the models scale to larger datasets, we train NODEs and ANODEs on 200 classes of $64 \times 64$ ImageNet. As can be seen in Fig. 13, ANODEs scale better, achieve lower losses and train almost 10 times faster than NODEs.

### 5.3 Relation to other models

In this section, we discuss how ANODEs compare to other relevant models.

**Neural ODEs.** Different architectures exist for training NODEs on images. Chen et al. (2018) for example, downsample MNIST twice with regular convolutions before applying a sequence of repeated ODE flows. These initial convolutions can be understood as implicitly augmenting the space (since they increase the number of channels). While combining NODEs with convolutions alleviates the representational limitations of NODEs, it also results in most of the attractive properties of NODEs being lost (including invertibility, ability to query state at any timestep, cheap Jacobian computations in normalizing flows and reduced number of parameters). In contrast, ANODEs overcome the representational weaknesses of NODEs while maintaining all their attractive properties.

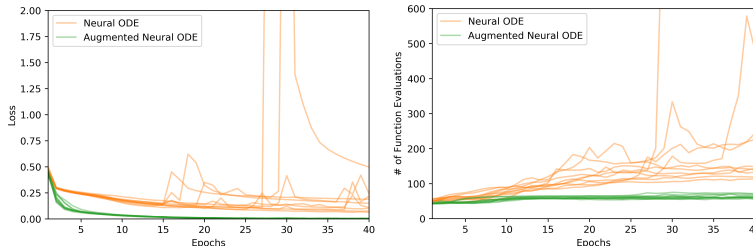

Figure 13: Instabilities in the loss (left) and NFEs (right) when fitting NODEs to MNIST. In the latter stages of training, NODEs can become unstable and the loss and NFEs become erratic.

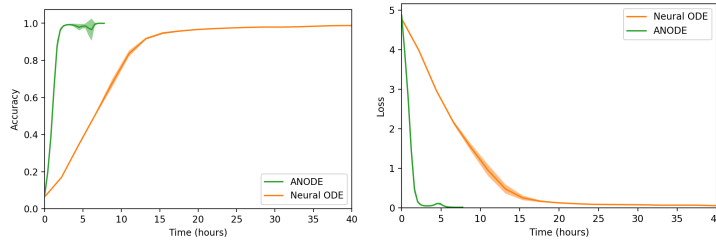

Figure 14: Accuracy (left) and loss (right) on $64 \times 64$ ImageNet for NODEs and ANODEs.

**ResNets.** Since ResNets can be interpreted as discretized equivalents of NODEs, it is interesting to consider how augmenting the space could affect the training of ResNets. Indeed, most ResNet architectures (He et al., 2016; Xie et al., 2017; Zagoruyko & Komodakis, 2016) already employ a form of augmentation by performing convolutions with a large number of filters before applying residual blocks. This effectively corresponds to augmenting the space by the number of filters minus the number of channels in the original image. Further, Behrmann et al. (2018) and Ardizzone et al. (2018) also augment the input with zeros to build invertible ResNets and transformations. Our findings in the continuous case are consistent with theirs: augmenting the input with zeros improves performance. However, an important consequence of using augmentation for NODEs is the reduced computational cost, which does not have an analogy in ResNets.

**Normalizing Flows.** Similarly to NODEs, several models used for normalizing flows, such as RealNVP (Dinh et al., 2016), MAF (Papamakarios et al., 2017) and Glow (Kingma & Dhariwal, 2018) are homeomorphisms. The results presented in this paper may therefore also be relevant in this context. In particular, using augmentation for discrete normalizing flows may improve performance and is an interesting avenue for future research.

## 6 Scope and Future Work

In this section, we describe some limitations of ANODEs, outline potential ways they may be overcome and list ideas for future work. First, while ANODEs are faster than NODEs, they are still slower than ResNets. Second, augmentation changes the dimension of the input space which, depending on the application, may not be desirable. Finally, the augmented dimension can be seen as an extra hyperparameter to tune. While the model is robust for a range of augmented dimensions, we observed that for excessively large augmented dimensions (e.g. adding 100 channels to MNIST), the model tends to perform worse with higher losses and NFEs. We believe the ideas presented in this paper could create interesting avenues for future research, including:

**Overcoming the limitations of NODEs.** In order to allow trajectories to travel across each other, we augmented the space on which the ODE is solved. However, there may be other ways to achieve this, such as learning an augmentation (as in ResNets) or adding noise (similarly to Wang et al. (2018)).

**Augmentation for Normalizing Flows.** The NFEs typically becomes prohibitively large when training continuous normalizing flow (CNF) models (Grathwohl et al., 2018). Adding augmentation to CNFs could likely mitigate this effect and we plan to explore this in future work.

**Improved understanding of augmentation.** It would be useful to provide more theoretical analysis for how and why augmentation improves the training of NODEs and to explore how this could guide our choice of architectures and optimizers for NODEs.

## 7 Conclusion

In this paper, we highlighted and analysed some of the limitations of Neural ODEs. We proved that there are classes of functions NODEs cannot represent and, in particular, that NODEs only learn features that are homeomorphic to the input space. We showed through experiments that this lead to slower learning and complex flows which are expensive to compute. To mitigate these issues, we proposed Augmented Neural ODEs which learn the flow from input to features in an augmented space. Our experiments show that ANODEs can model more complex functions using simpler flows while achieving lower losses, reducing computational cost, and improving stability and generalization.

## Acknowledgements

We would like to thank Anthony Caterini, Daniel Paulin, Abraham Ng, Joost Van Amersfoort and Hyunjik Kim for helpful discussions and feedback. Emilien gratefully acknowledges his PhD funding from Google DeepMind. Arnaud Doucet acknowledges support of the UK Defence Science and Technology Laboratory (Dstl) and Engineering and Physical Research Council (EPSRC) under grant EP/R013616/1. This is part of the collaboration between US DOD, UK MOD and UK EPSRC under the Multidisciplinary University Research Initiative. Yee Whye Teh's research leading to these results has received funding from the European Research Council under the European Union's Seventh Framework Programme (FP7/2007-2013) ERC grant agreement no. 617071.

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
