[Supplementary Material]

# A  Proofs

Throughout this section, we refer to the following Initial Value Problem (IVP)

$$\begin{cases} \dfrac{\mathrm{d}\mathbf{h}(t)}{\mathrm{d}t} = \mathbf{f}(\mathbf{h}(t), t) \\ \mathbf{h}(0) = \mathbf{x} \end{cases} \tag{1}$$

where $\mathbf{h}(t) \in \mathbb{R}^d$ and $\mathbf{f} : \mathbb{R}^d \times \mathbb{R} \to \mathbb{R}^d$ is continuous in $t$ and globally Lipschitz continuous in $\mathbf{h}$, i.e. there is a constant $C \geq 0$ such that

$$\|\mathbf{f}(\mathbf{h}_2(t), t) - \mathbf{f}(\mathbf{h}_1(t), t)\| \leq C\|\mathbf{h}_2(t) - \mathbf{h}_1(t)\|$$

for all $t \in \mathbb{R}$. These conditions imply the solutions of the IVP exist and are unique for all $t$ (see e.g. Theorem 2.4.5 in Ahmad & Ambrosetti (2015)).

We define the flow $\phi_t(\mathbf{x})$ associated to the vector field $\mathbf{f}(\mathbf{h}(t), t)$ as the solution at time $t$ of the ODE starting from the initial condition $\mathbf{h}(0) = \mathbf{x}$. The flow measures how the solutions of the ODE depend on the initial conditions. Following the analogy between ResNets and NODEs, we define the features $\phi(\mathbf{x})$ output by the ODE as the flow at the final time $T$ to which we solve the ODE, i.e. $\phi(\mathbf{x}) = \phi_T(\mathbf{x})$. Finally, we define the NODE model as the composition of the feature function $\phi : \mathbb{R}^d \to \mathbb{R}^d$ and a linear map $\mathcal{L} : \mathbb{R}^d \to \mathbb{R}$.

For clarity and completeness, we include proofs of all statements. Whenever propositions or theorems are already known we include references to proofs.

## A.1  ODE trajectories do not intersect

This result is well known and proofs can be found in standard ODE textbooks (e.g. Proposition C.6 in Younes (2010)).

**Proposition.** *Let $\mathbf{h}_1(t)$ and $\mathbf{h}_2(t)$ be two solutions of the ODE (1) with different initial conditions, i.e. $\mathbf{h}_1(0) \neq \mathbf{h}_2(0)$. Then, for all $t \in (0, T]$, $\mathbf{h}_1(t) \neq \mathbf{h}_2(t)$. Informally, this proposition states that ODE trajectories cannot intersect.*

*Proof.* Suppose there exists some $\tilde{t} \in (0, T]$ where $\mathbf{h}_1(\tilde{t}) = \mathbf{h}_2(\tilde{t})$. Define a new IVP with initial condition $\mathbf{h}(\tilde{t}) = \mathbf{h}_1(\tilde{t}) = \mathbf{h}_2(\tilde{t})$ and solve it backwards to time $t = 0$. As the backwards IVP also satisfies the existence and uniqueness conditions, its solution $\mathbf{h}(t)$ is unique implying that its value at $t = 0$ is unique. This contradicts the assumption that $\mathbf{h}_1(0) \neq \mathbf{h}_2(0)$ and so there is no $\tilde{t} \in (0, T]$ such that $\mathbf{h}_1(\tilde{t}) = \mathbf{h}_2(\tilde{t})$.

## A.2  Gronwall's Lemma

We will make use of Gronwall's Lemma and state it here for completeness. We follow the statement as given in Howard (1998):

**Theorem.** *Let $U \subset \mathbb{R}^d$ be an open set. Let $\mathbf{f} : U \times [0, T] \to \mathbb{R}^d$ be a continuous function and let $\mathbf{h}_1, \mathbf{h}_2 : [0, T] \to U$ satisfy the IVPs:*

$$\frac{\mathrm{d}\mathbf{h}_1(t)}{\mathrm{d}t} = \mathbf{f}(\mathbf{h}_1(t), t), \quad \mathbf{h}_1(0) = \mathbf{x}_1$$

$$\frac{\mathrm{d}\mathbf{h}_2(t)}{\mathrm{d}t} = \mathbf{f}(\mathbf{h}_2(t), t), \quad \mathbf{h}_2(0) = \mathbf{x}_2$$

*Assume there is a constant $C \geq 0$ such that*

$$\|\mathbf{f}(\mathbf{h}_2(t), t) - \mathbf{f}(\mathbf{h}_1(t), t)\| \leq C\|\mathbf{h}_2(t) - \mathbf{h}_1(t)\|$$

*Then for $t \in [0, T]$*

$$\|\mathbf{h}_2(t) - \mathbf{h}_1(t)\| \leq e^{Ct}\|\mathbf{x}_2 - \mathbf{x}_1\|$$

*Proof.* See e.g. Howard (1998) or Theorem 3.8 in Younes (2010).

# B Proof for 1d example

Let $g_{1d} : \mathbb{R} \to \mathbb{R}$ be a function such that

$$\begin{cases} g_{1d}(-1) = 1 \\ g_{1d}(1) = -1 \end{cases}$$

**Proposition 1.** *The flow of an ODE cannot represent $g_{1d}(x)$.*

*Proof.* The proof follows two steps:

    (a) Continuous trajectories mapping $-1$ to $1$ and $1$ to $-1$ must cross each other.

    (b) Trajectories of ODEs cannot cross each other.

This is a contradiction and implies the proposition. Part (b) was proved in Section A.1. All there is left to do is to prove part (a).

Suppose there exists an $\mathbf{f}$ such that there are trajectories $h_1(t)$ and $h_2(t)$ where

$$\begin{cases} h_1(0) = -1 \quad h_1(T) = 1 \\ h_2(0) = 1 \quad h_2(T) = -1 \end{cases}$$

As $h_1(t)$ and $h_2(t)$ are solutions of the IVP, they are continuous; see, e.g., Coddington & Levinson (1955). Define the function $h(t) = h_2(t) - h_1(t)$. Since both $h_1(t)$ and $h_2(t)$ are continuous, so is $h(t)$. Now $h(0) = 2$ and $h(T) = -2$, so by the Intermediate Value Theorem there is some $\tilde{t} \in [0, T]$ where $h(\tilde{t}) = 0$, i.e. where $h_1(\tilde{t}) = h_2(\tilde{t})$. So $h_1(t)$ and $h_2(t)$ intersect.

# C Proof that $\phi_t(\mathbf{x})$ is a homeomorphism

Since the following theorem plays a central part in the paper, we include a proof of it here for completeness. For a more general proof, we refer the reader to Theorem C.7 in Younes (2010).

**Theorem.** *For all $t \in [0, T]$, $\phi_t : \mathbb{R}^d \to \mathbb{R}^d$ is a homeomorphism.*

*Proof.* In order to prove that $\phi_t$ is a homemorphism, we need to show that

    (a) $\phi_t$ is continuous

    (b) $\phi_t$ is a bijection

    (c) $\phi_t^{-1}$ is continuous

*Part (a).* Consider two initial conditions of the ODE system, $\mathbf{h}_1(0) = \mathbf{x}$ and $\mathbf{h}_2(0) = \mathbf{x} + \delta$ where $\delta$ is some perturbation. By Gronwall's Lemma, we have

$$\|\mathbf{h}_2(t) - \mathbf{h}_1(t)\| \le e^{Ct}\|\mathbf{h}_1(0) - \mathbf{h}_2(0)\| = e^{Ct}\|\delta\|$$

Rewriting in terms of $\phi_t(\mathbf{x})$, we have

$$\|\phi_t(\mathbf{x} + \delta) - \phi_t(\mathbf{x})\| \le e^{Ct}\|\delta\|$$

Letting $\delta \to 0$, this implies that $\phi_t(\mathbf{x})$ is continuous in $\mathbf{x}$ for all $t \in [0, T]$.

*Part (b).* Suppose there exists initial conditions $\mathbf{x}_1 \ne \mathbf{x}_2$ such that $\phi_t(\mathbf{x}_1) = \phi_t(\mathbf{x}_2)$. We define the IVP starting from $\phi_t(\mathbf{x}_1)$ and solve it backwards to time $t = 0$. The solution of the IVP is unique, so it cannot map $\phi_t(\mathbf{x}_1)$ back to both $\mathbf{x}_1$ and $\mathbf{x}_2$. So for each $\mathbf{x}_1 \ne \mathbf{x}_2$, we must have $\phi_t(\mathbf{x}_1) \ne \phi_t(\mathbf{x}_2)$, that is the map between $\mathbf{x}$ and $\phi_t(\mathbf{x})$ is one-to-one.

*Part (c).* To check that the inverse $\phi_t^{-1}$ is continuous, we note that we can set the initial condition to $\mathbf{h}(t) = \phi_t(\mathbf{x})$ and solve the IVP backwards in time (as it satisfies the existence and uniqueness conditions). The same reasoning as part (a) then applies.

Therefore $\phi_t$ is a continuous bijection and its inverse is continuous, i.e. it is a homeomorphism.

Figure 1: (a) Diagram of $g(\mathbf{x})$ in 2d. (b) An example of the map $\phi(\mathbf{x})$ from input data to features necessary to represent $g(\mathbf{x})$ (which NODEs cannot learn).

**Corollary.** *Features of Neural ODEs preserve the topology of the input space.*

*Proof.* Since $\phi_t(\mathbf{x})$ is a homeomorphism, so is $\phi(\mathbf{x}) = \phi_T(\mathbf{x})$. Homeomorphims preserve topological properties, so Neural ODEs can only learn features which have the same topology as the input space.

This corollary implies for example that NODEs cannot break apart or create holes in a connected region of the input space.

# D  Proof that there are classes functions NODEs cannot represent

This section presents a proof of the main claim of the paper.

Let $0 < r_1 < r_2 < r_3$ and let $g : \mathbb{R}^d \to \mathbb{R}$ be a function such that

$$\begin{cases} g(\mathbf{x}) = -1 & \text{if } \|\mathbf{x}\| \leq r_1 \\ g(\mathbf{x}) = 1 & \text{if } r_2 \leq \|\mathbf{x}\| \leq r_3 \end{cases}$$

We denote the sphere where $g(\mathbf{x}) = -1$ by $A = \{\mathbf{x} : \|\mathbf{x}\| \leq r_1\}$ and the annulus where $g(\mathbf{x}) = 1$ by $B = \{\mathbf{x} : r_2 \leq \|\mathbf{x}\| \leq r_3\}$ (see Fig. 1). For a set $S$, we write $\phi(S) = \{\mathbf{y} : \mathbf{y} = \phi(\mathbf{x}), \mathbf{x} \in S\}$ to denote the feature transformation of the set.

**Proposition 2.** *Neural ODEs cannot represent $g(\mathbf{x})$.*

*Proof.* For a NODE to map points in $A$ to $-1$ and points in $B$ to $+1$, the linear map $\mathcal{L}$ must map the features in $\phi(A)$ to $-1$ and the features in $\phi(B)$ to $+1$, which implies that $\phi(A)$ and $\phi(B)$ must be linearly separable. We now show that this is not possible if $\phi$ is a homeomorphism.

Define a disk $D \subset \mathbb{R}^d$ by $D = \{\mathbf{x} \in \mathbb{R}^d : \|\mathbf{x}\| \leq r_2\}$ with boundary $\partial D = \{\mathbf{x} \in \mathbb{R}^d : \|\mathbf{x}\| = r_2\}$ and interior $\text{int}(D) = \{\mathbf{x} \in \mathbb{R}^d : \|\mathbf{x}\| < r_2\}$. Now $A \subset \text{int}(D)$, $A \cap \partial D = \emptyset$ and $\partial D \subset B$, that is all points in $\partial D$ should be mapped to $+1$ (i.e. they are in $B$) and a subset of points in $\text{int}(D)$ should be mapped to $-1$ (i.e. they are in $A$). So if $\phi(\text{int}(D))$ and $\phi(\partial D)$ are not linearly separable, then neither are $\phi(A)$ or $\phi(B)$.

The feature transformation $\phi$ is a homeomorphism, so $\phi(\text{int}(D)) = \text{int}(\phi(D))$ and $\phi(\partial D) = \partial(\phi(D))$, i.e. points on the boundary get mapped to points on the boundary and points in the interior to points in the interior (Armstrong, 2013). So it remains to show that $\text{int}(\phi(D))$ and $\partial(\phi(D))$ cannot be linearly separated. For notational convenience, we will write $D' = \phi(D)$.

Suppose all points in $\partial D'$ lie above some hyperplane, i.e. suppose there exists a linear function $\mathcal{L}(\mathbf{x}) = \mathbf{w}^T \mathbf{x}$ and a constant $C$ such that $\mathcal{L}(\mathbf{x}) > C$ for all $\mathbf{x} \in \partial D'$. If $\text{int}(D')$ were linearly separable from $\partial D'$ then $\mathcal{L}(\mathbf{x}) < C$ for all $\mathbf{x} \in \text{int}(D')$. We now show that this is not the case. Since $D'$ is a connected subset of $\mathbb{R}^d$ (since $D$ is connected and $\phi$ is a homeomorphism), every point $\mathbf{x} \in \text{int}(D')$ can be written as a convex combination of points on the boundary $\partial D'$ (to see this consider a line passing through a point $\mathbf{x}$ in the interior and its intersection with the boundary). So if $\mathbf{x} \in \text{int}(D')$, then

$$\mathbf{x} = \lambda \mathbf{x}_1 + (1 - \lambda)\mathbf{x}_2$$

for some $\mathbf{x}_1, \mathbf{x}_2 \in \partial D'$ and $0 < \lambda < 1$. Now,

$$
\begin{aligned}
\mathcal{L}(\mathbf{x}) &= \mathbf{w}^T \mathbf{x} \\
&= \mathbf{w}^T (\lambda \mathbf{x}_1 + (1 - \lambda)\mathbf{x}_2) \\
&= \lambda \mathbf{w}^T \mathbf{x}_1 + (1 - \lambda)\mathbf{w}^T \mathbf{x}_2 \\
&\geq \lambda C + (1 - \lambda)C \\
&= C
\end{aligned}
$$

so all points in the interior are on the same side of the hyperplane as points on the boundary, that is the interior and the boundary are not linearly separable. This implies that the set of features $\phi(A)$ and $\phi(B)$ cannot be linearly separated and so that NODEs cannot represent $g(\mathbf{x})$.

Figure 2: (a) Diagram of the disk $D$ and its boundary. The boundary is equal to the inner boundary of $B$. (b) An example of how $\phi$ transforms the disk. (c) The boundary of the transformed set is above the hyperplane, which implies that all points on the interior must also be above the hyperplane.

# E    Modeling NODEs and $\mathbf{f}(\mathbf{h}(t), t)$

In this section, we describe how to choose and model $\mathbf{f}$. We first note that $\mathbf{f}$ can be parameterized by any standard neural net architecture, including ones with activation functions that are not everywhere differentiable such as ReLU. Existence and uniqueness of solutions to the ODE are still guaranteed and all results in this paper hold under these conditions.

The function $\mathbf{f}(\mathbf{h}(t), t)$ depends on both the time $t$ and the hidden state $\mathbf{h}(t)$. Following the architecture used by Chen et al. (2018), we model $\mathbf{f}$ as a CNN or an MLP with weights that are not a function of time, and instead encode the time dependency by passing a concatenated tensor $(\mathbf{h}(t), t)$ as input to the neural network. The architectures of the CNNs and MLPs we used are described in the following section.

# F    Experimental Details

We used the ODE solvers in the `torchdiffeq`[1] library for all experiments (Chen et al., 2018). We used the adaptive Dormand-Prince (Runge-Kutta 45) solver with an absolute and relative error tolerance of 1e-3. The code to reproduce all results in this paper can be found at `https://github.com/EmilienDupont/augmented-neural-odes`.

## F.1    Architecture

Throughout all our experiments we used the ReLU activation function. We also experimented with softplus but found that this generally slowed down learning.

### F.1.1 Toy datasets

We parameterized **f** by an MLP with the following structure and dimensions

$$d_{\text{input}} + 1 \rightarrow d_{\text{hidden}} \rightarrow \text{ReLU} \rightarrow d_{\text{hidden}} \rightarrow \text{ReLU} \rightarrow d_{\text{input}}$$

where the additional dimension on the input layer is because we append the time $t$ as an input. Choices for $d_{\text{input}}$ and $d_{\text{hidden}}$ are given for each model in the following section.

### F.1.2 Image datasets

We parameterized **f** by a convolutional block with the following structure and dimensions

- $1 \times 1$ conv, $k$ filters, 0 padding.
- $3 \times 3$ conv, $k$ filters, 1 padding.
- $1 \times 1$ conv, $c$ filters, 0 padding.

where $k$ is specified for each architecture in the following sections and $c$ is the number of channels (1 for MNIST and 3 for CIFAR10, SVHN and ImageNet). We append the time $t$ as an extra channel on the feature map before each convolution.

## F.2 Hyperparameters

For the toy datasets, each experiment was repeated 20 times. The resulting plots show the mean and standard deviation for these runs.

### F.2.1 Hyperparameter search

To ensure a fair comparison between models, we ran a large hyperparameter search for each model and chose the hyperparameters with the lowest loss to generate the plots in the paper. We used `skorch` and `scikit-learn` (Pedregosa et al., 2011) to run the hyperparameter searches and ran 3 cross validations for each setting.

For $d = 1$ and $d = 2$ we trained on $g(\mathbf{x})$ (i.e. on the dataset of concentric spheres), with 1000 points in the inner sphere and 2000 points in the outer annulus. We used $r_1 = 0.5$, $r_2 = 1.0$ and $r_3 = 1.5$ and trained for 50 epochs. The space of hyperparameters we searched were:

- Batch size: 64, 128
- Learning rate: 1e-3, 5-4, 1e-4
- Hidden dimension: 16, 32
- Number of layers (for ResNet): 2, 5, 10
- Number of augmented dimensions (for ANODE): 1, 2, 5

The best parameters for ResNets:

- $d = 1$: Batch size 64, learning rate 1e-3, hidden dimension 32, 5 layers
- $d = 2$: Batch size 64, learning rate 1e-3, hidden dimension 32, 5 layers

The best parameters for Neural ODEs:

- $d = 1$: Batch size 64, learning rate 1e-3, hidden dimension 32
- $d = 2$: Batch size 64, learning rate 1e-3, hidden dimension 32

The best parameters for Augmented Neural ODEs:

- $d = 1$: Batch size 64, learning rate 1e-3, hidden dimension 32, augmented dimension 5
- $d = 2$: Batch size 64, learning rate 1e-3, hidden dimension 32, augmented dimension 5

### F.2.2 Image experiments

For all image datasets, we used $k = 64$ filters and repeated each experiment 5 times. For models with approximately the same number of parameters we used, for MNIST

- NODE: 92 filters $\rightarrow$ 84,395 parameters
- ANODE: 64 filters, augmented dimension 5 $\rightarrow$ 84,816 parameters

and for CIFAR10 and SVHN

- NODE: 125 filters $\rightarrow$ 172,358 parameters
- ANODE: 64 filters, augmented dimension 10 $\rightarrow$ 171,799 parameters

For the ImageNet experiments, we used the `Tiny ImageNet` dataset consisting of 200 classes of $64 \times 64$ images. We also repeated each experiment 5 times. We used models with approximately the same number of parameters, specifically:

- NODE: 164 filters $\rightarrow$ 366,269 parameters
- ANODE: 64 filters, augmented dimension 5 $\rightarrow$ 365,714 parameters

For all image experiments, we used a batch size of 256.

## G    Additional Results

In this section, we show additional results which were not included in the main paper.

### G.1    Feature space evolution

We visualize the evolution of the feature space when training a NODE on $g(\mathbf{x})$ and on a separable function in Fig. 3. As can be seen, the NODE struggles to push the inner sphere out of the annulus for $g(\mathbf{x})$. On the other hand, when training on the separable dataset, the NODE easily transforms the input space.

### G.2    Parameter efficiency

As noted in the main paper, when we augment the dimension of the ODEs, we also increase the number of parameters of the model. We test whether the improved performance of ANODEs is due to the higher number of parameters by training NODEs and ANODEs with the same number of parameters on MNIST and CIFAR10. As can be seen in Fig. 4, the augmented model achieves lower losses with fewer NFEs than a NODE with the same number of parameters, suggesting that ANODEs use the parameters more efficiently than NODEs.

### G.3    Augmentation and weight decay

Grathwohl et al. (2018) train NODE models with weight decay to reduce the NFEs. As ANODEs also achieve low NFEs, we test models with various combinations of weight decay and augmentation and show results in Fig. 5. We find that ANODEs significantly outperform NODEs even when using weight decay. However, using both weight decay and augmentation achieves the lowest NFEs at the cost of a slightly higher loss.

### G.4    Comparing ResNets, NODEs and ANODEs

In the main paper, we compare the training time of ResNets with NODEs and the training time of NODEs with ANODEs. In Fig. 6, we compare all three methods in a single plot.

### G.5    Training accuracy

We include additional plots of training accuracy in Fig. 7.

Figure 3: Evolution of the feature space during training. The leftmost tile shows the feature space for a randomly initialized NODE and the rightmost tile shows the feature space after training. The top row shows a model trained on $g(\mathbf{x})$ and the bottow row a model trained on a separable function.

Figure 4: Losses, NFEs and NFEs vs Loss for various augmented models on MNIST and CIFAR10. Note that $p$ indicates the size of the augmented dimension, so $p = 0$ corresponds to a regular NODE model.

## G.6 Additional test results for SVHN

The test loss and accuracy on SVHN as training progresses are shown in Fig. 8.

## G.7 Additional train results for SVHN

Additional results for SVHN which were not included in the main paper are shown in Fig. 9.

## G.8 Additional results for ImageNet

Additional results for ImageNet which were not included in the main paper are shown in Fig. 10.

## G.9 Examples of flows

We include further plots of flows learned by NODEs and ANODEs in Fig. 11. As can be seen, ANODEs consistently learn simple, nearly linear flows, while NODEs require more complicated flows to separate the data.

Figure 5: Losses and NFEs for models with and without weight decay. ANODEs perform better than NODEs with weight decay but adding weight decay to ANODEs also reduces their NFEs at the cost of a slightly higher loss.

Figure 6: Losses for various models trained on $g(\mathbf{x})$ in $d = 2$. As can be seen, ANODEs are slightly slower than ResNets, but faster than NODEs.

Figure 7: Training accuracy for MNIST (left), CIFAR10 (middle) and SVHN (right).

Figure 8: Test loss and accuracy during training for SVHN.

Figure 9: Loss, NFEs, loss vs NFEs during training for NODEs and ANODEs on SVHN.

Figure 10: NFEs, loss vs NFEs and accuracy vs NFEs during training for NODEs and ANODEs on $64 \times 64$ ImageNet.

Figure 11: Flows learned by NODEs and ANODEs trained on various datasets. The top row shows results for NODEs, the bottom row shows results for ANODEs. The models in the left column were trained on separable data, whereas the models in the right column were trained on $g(\mathbf{x})$. NODEs learn more complex flows, particularly on data which is not separable.

## Footnotes

[1] https://github.com/rtqichen/torchdiffeq