[Reviews · NeurIPS 2019]

Reviewer 1



The paper is very well written, however, there are several questions about novelty of the work detailed below

Reviewer 2



- Originality To the best of my knowledge, this is a novel work. The representational power of neural ODE models has not been studied much in the field. Although the negative examples and proof techniques are standard results in point-set topology and metric spaces, the appropriate application makes the idea very interesting. The related works section seems adequate. - Quality The work is technically sound, with both cleanly written proofs and comprehensive empirical analysis. - Clarity The work is clear, concise, and coherent. - Significance See part 1 of the review.

Reviewer 3



Originality: The method is original in the deep learning literature. Though limitations of ODEs cannot cross paths is quite well-known, this paper views this deficiency from a modeling perspective and removes it while keeping within the ODE framework. Quality & Clarity: The motivations for ANODE are well-explained and the experiments are well-chosen. The prose is very well written, and with many simple visualizations that support their claims. Significance: Given the interest in ODE-based modeling, this work has enough impact for a NeurIPS paper. Comments: - I think reporting _only_ cross entropy loss for image classification tasks is a bit weird. If it makes sense to compare cross-entropy because it is objective that is being minimized for training (e.g. training instability plots), sure. But I think showing classification accuracy would be more meaningful and allow more follow-up works as it is the metric of interest. It is often the case in image classification that while validation cross entropy increases, the classification actually gets better. Right now, Figure 11 makes it seem like ANODE has a more significant overfitting problem, though I think the accuracy probably shouldn't increase very much even if the loss increases.

[Author Response · NeurIPS 2019]

We would like to thank the reviewers for their clear and thorough reviews. In addition to the positive feedback, we are also grateful for the valuable comments and suggestions for improving the paper. Below, we answer the reviewers' questions and discuss how we will incorporate the suggested improvements and changes.

**Classification accuracy.** We agree with the reviewers that reporting classification accuracy is important. We have updated the paper to include this information and confirm that both train and test accuracy are significantly improved when using Augmented Neural ODEs (ANODEs) on all the datasets we have considered. For example, test accuracy improves by 2% on MNIST and by 6% on CIFAR10. We also note that ANODEs achieve these accuracies in about $5\times$ fewer iterations than Neural ODEs (NODEs).

**Connection to known ODE results.** As mentioned by the reviewers, the fact that ODE trajectories cannot cross and the fact that ODE flows are homeomorphisms are quite well known results. However, to the best of our knowledge, and in agreement with Reviewers 2 and 3, interpreting these results from a modeling perspective and showing the connection between them and the representational power of NODEs is a novel contribution. Further, the consequences of these results, including the increased computational cost, higher losses and poorer generalization of NODEs are also, to the best of our knowledge, original contributions. In addition, the empirical results showing that the representational limitations make a considerable difference in practice are also novel. However, we agree with Reviewer 1 that we could have made the novelty our of results clearer and we will update the paper to describe our contributions in more detail.

**Combining NODEs with convolutional layers.** As noted by Reviewer 1, since regular convolutional (non-ODE) layers do not have the representational limitations of ODE layers, combining them with ODE layers results in models that (in a similar way to ANODEs) can represent functions NODEs cannot. However, combining NODEs with standard convolutional layers also means that most of the attractive properties of NODEs are lost (including invertibility, ability to query hidden state at any timestep, cheap Jacobian computations in normalizing flows and reduced number of parameters). In contrast, ANODEs overcome the representational weaknesses of NODEs while maintaining all their attractive properties. Indeed, identifying and overcoming the representational limitations and slow computation of NODEs, while maintaining all the advantages of an ODE framework is, in our eyes, one of the main contributions of the paper.

**Augmentation in ResNets.** As described in the paper and as noted by Reviewer 1, augmentation by adding more channels has been used in the context of ResNets (e.g. in Wide Residual Networks). However, these changes were empirically motivated and we believe our work provides a novel perspective on why using wider networks may be useful in the context of ResNets. More importantly, in the case of NODEs, which are the main focus of our work, we are not aware of any work using augmentation. Further, an important consequence of using augmentation for NODEs is the reduced computational cost, which does not have an analogy in ResNets. We will add a more detailed discussion of this. We also thank Reviewer 1 for pointing us to Wide Residual Networks which we will cite in the updated paper.

**Number of parameters in ANODEs.** We welcome the feedback from Reviewer 1 on including more results comparing NODEs and ANODEs with the same number of parameters. In all our experiments, we found that ANODEs consistently and significantly outperform NODEs even with the same number of parameters. We will update the paper so every comparison between NODEs and ANODEs is done on models with the same number of parameters.

**Generalization.** We confirm that the improved generalization of ANODEs holds not only for CIFAR10 (which was included in the paper) but also for all the other datasets we have considered. This is true both in terms of test loss and test accuracy. We will include additional results on this in the updated paper. To respond to Reviewer 1's question on overfitting, we note that while ANODEs get better test accuracy than NODEs, they also tend to overfit the data more. This is because ANODEs are more powerful function approximators and are (empirically) easier to optimize. Using standard regularization techniques mitigates this effect. We will update the paper to include a discussion of this.

**Augmented channels.** We agree with Reviewer 2 that using the expression augmented channels is more appropriate than augmented dimensions (since there are indeed $10hw$ augmented dimensions in the image case) and we will update the paper to reflect this.

**Forward and backward NFEs.** We confirm that both forward and backward NFEs are reduced when using ANODEs. We will include plots of this in the updated paper and thank Reviewer 2 for pointing this out.

**Notation for functions.** We agree with Reviewer 2 that using the notation $\phi_{1d}(x)$ instead of $g_{1d}(x)$ in Section 3 is more appropriate, since this function is indeed a flow.

**Bigger datasets.** We welcome the feedback from Reviewer 3 on running experiments on more datasets and will include results on SVHN (in addition to the already included MNIST, CIFAR10 and tiny ImageNet) in the updated paper.

We would like to thank the reviewers again for their positive comments and feedback. We are excited to incorporate the suggested changes which will definitely improve the quality of the paper.

[Meta-Review · NeurIPS 2019]

This paper connects a well-known result about the limits of diffeomorphisms, and applies it to the recent neural ODE model. The authors do experiments to show how adding extra channels reduces the computational cost of these models as well. R1 makes the valid point that the theoretical result was shown in 1955, and that the engineering trick of making layers wider is resnets existed previously. However, I'd say that the main contribution of this paper is in connecting these ideas to neural ODEs, and giving a possible explanation of why wider layers help in resnets. This paper also pushes forward our practical understanding of training neural ODEs. It's a clear story, and well-written paper. However, the paper and rebuttal avoided reporting absolute (probably poor) classification results. It's unclear if investigating neural ODEs for the classification problems considered here is a pressing direction. There are other uses for neural ODEs besides classification, but it's not clear that we can add extra dimensions in those settings without losing other good properties. For example, the paper and rebuttal mention normalizing flows, and one of the positive reviewers thought that would make sense as an application. However, it's not immediately clear how to apply augmentation to flows: one would no longer immediately get the probability of the data, the main attraction of flows. We encourage the authors to update the camera-ready to be upfront about any limitations in the classification performance of the architectures explored.